# AgTx2-GFP, Fluorescent Blocker Targeting Pharmacologically Important K_v_1.x (x = 1, 3, 6) Channels

**DOI:** 10.3390/toxins15030229

**Published:** 2023-03-18

**Authors:** Alexandra L. Primak, Nikita A. Orlov, Steve Peigneur, Jan Tytgat, Anastasia A. Ignatova, Kristina R. Denisova, Sergey A. Yakimov, Mikhail P. Kirpichnikov, Oksana V. Nekrasova, Alexey V. Feofanov

**Affiliations:** 1Shemyakin-Ovchinnikov Institute of Bioorganic Chemistry, Russian Academy of Sciences, 117997 Moscow, Russia; 2Faculty of Biology, Lomonosov Moscow State University, 119234 Moscow, Russia; 3Toxicology and Pharmacology, Campus Gasthuisberg O&N2, University of Leuven (KU Leuven), Herestraat 49, P.O. Box 922, B-3000 Leuven, Belgium

**Keywords:** agitoxin 2, GFP, voltage-gated K_v_1 channels, KcsA-K_v_1 channels, affinity, confocal, patch clamp, oocytes, Neuro 2a cells

## Abstract

The growing interest in potassium channels as pharmacological targets has stimulated the development of their fluorescent ligands (including genetically encoded peptide toxins fused with fluorescent proteins) for analytical and imaging applications. We report on the properties of agitoxin 2 C-terminally fused with enhanced GFP (AgTx2-GFP) as one of the most active genetically encoded fluorescent ligands of potassium voltage-gated K_v_1.x (x = 1, 3, 6) channels. AgTx2-GFP possesses subnanomolar affinities for hybrid KcsA-K_v_1.x (x = 3, 6) channels and a low nanomolar affinity to KcsA-K_v_1.1 with moderate dependence on pH in the 7.0–8.0 range. Electrophysiological studies on oocytes showed a pore-blocking activity of AgTx2-GFP at low nanomolar concentrations for K_v_1.x (x = 1, 3, 6) channels and at micromolar concentrations for K_v_1.2. AgTx2-GFP bound to K_v_1.3 at the membranes of mammalian cells with a dissociation constant of 3.4 ± 0.8 nM, providing fluorescent imaging of the channel membranous distribution, and this binding depended weakly on the channel state (open or closed). AgTx2-GFP can be used in combination with hybrid KcsA-K_v_1.x (x = 1, 3, 6) channels on the membranes of *E. coli* spheroplasts or with K_v_1.3 channels on the membranes of mammalian cells for the search and study of nonlabeled peptide pore blockers, including measurement of their affinity.

## 1. Introduction

Many peptide toxins from scorpion venoms are potent and specific blockers of potassium channels [1,2], which are tetrameric transmembrane proteins with a central ion-conducting pore [3]. The toxins bind at the outer vestibule of a channel, occluding the pore with a lysine residue. The main targets of scorpion peptide toxins are voltage-gated K_v_1.x channels, calcium-activated K^+^ channels (K_Ca_), and K_v_11.x channels [1,2,4]. The affinity profile of each peptide toxin usually comprises a limited set of K^+^-channel isoforms, suggesting that scorpion toxins are able to discriminate a few targets from dozens of different K^+^ channels [1,5]. For example, the OSK1 toxin from the scorpion *Orthochirus scrobiculosus* is a high-affinity blocker of K_v_1.1 and K_v_1.3 channels [6], is moderately active on K_v_1.2 [6], is weakly active on K_v_1.6 and K_Ca_3.1 channels [6,7] and is not active on K_v_1.4, K_v_1.5, K_v_1.7, K_v_3.1, K_v_3.2, K_v_11, K_Ca_11, K_Ca_2.1, K_Ca_2.2, and K_Ca_2.3 channels [6,7,8]. The BeKm1 toxin from the scorpion *Mesobuthus eupeus* is a potent blocker of the K_v_11.1, K_v_11.2, K_v_11.3, and K_v_12.1 channels [9] and is not active on the K_v_2.3, K_v_7.1, K_v_7.2, K_v_7.4, K_v_10, K_Ca_1.1, K_Ca_2.1, K_Ca_2.2, and K_Ca_3.1 channels [6,7,8]. Some scorpion toxins are highly selective, showing high-affinity binding to a single target channel, such as, for example, the K_v_1.3 channel blockers agitoxin 1 and Vm24 [10,11] or the K_v_1.2 channel blocker MeKTx11-1 [12].

The properties of scorpion toxins have made them a unique tool for studying K^+^ channels, which are abundant in almost all types of cells and tissues [6,13,14]. The structure and functions of K^+^ channels have been clarified with such peptides as margatoxin, charybdotoxin, agitoxin 2 (AgTx2), iberiotoxin, scyllatoxin, Vm24, HsTX1[R14A], and some others [1,13,14,15,16,17,18]. Charybdotoxin, a potent blocker of several K_v_1 and K_Ca_ channels, has been used to study structural determinants of toxin–channel interactions [19,20]. AgTx2 and margatoxin helped to prove the involvement of K_v_1.1 and K_v_1.3 channels in the mechanisms of retinal ganglion cell degeneration [21] and to identify the expression of K_v_1.3 in activated microglial cells as a probable reason for the modulation of microglia proliferation [22]. When applying the K_v_1.3 channel peptide blocker HsTX1[R14A] to microglial cells, it was found that the levels of TNF-alpha, IL-6, and nitric oxide production associated with microglia-mediated neuroinflammation depended on the K_v_1.3 channel activity [18]. The Vm24 toxin assisted in a study of T_EM_ cell response to K_v_1.3 blocking [17]. By using iberiotoxin, it was demonstrated that blocking the K_Ca_1.1 channel enhanced cell migration, thus indicating the role of the K_Ca_1.1 channel in intestinal epithelial restitution [15]. The compounds and mechanisms modulating the expressions of small-conductance Ca^2+^-activated K^+^ channels in human neuroblastoma cells were studied using scyllatoxin [16].

The growing interest in potassium channels as pharmacological targets associated with autoimmune, cancer, and neurological diseases [23,24,25,26,27,28,29] has provoked a major effort in using peptide toxins as a pharmacophore for engineering advanced peptides with increased selectivity and affinity to target channels [30,31,32,33]. It was shown that therapy-oriented selective peptide blockers can be developed using phage display based on a library of scorpion toxin sequences [34]. Moreover, this interest has stimulated the development and refinement of a wide variety of labeled blockers for different analytical and imaging applications. Radio- or fluorescently labeled peptide blocker derivatives are successfully used in radioligand or fluorescence-based binding assays, as well as in studies on K^+^ channel traffic and distribution [35].

In many cases, the labeling strategy includes the construction of a mutated peptide (by inserting Tyr or Cys residue) with retained biological activity, followed by the iodination of Tyr with ^125^I [36,37], and the labeling of Cys with [^3^H]-N-ethylmaleimide [38] or with thiol-reactive fluorescent dyes [39,40]. Other approaches are based on the labeling of the N-terminal amino group with an organic fluorophore during solid-phase synthesis of the peptide [41,42,43] or the “blind” labeling of one of several Lys residues, which are present in the peptide, with an amino-reactive organic fluorophore [44]. In the latter case, the separation of labeled derivatives and the selection of the most active ones followed by the determination of the modified Lys residue are required.

An alternative strategy to create fluorescently labeled peptide blockers is the production of genetically encoded peptide toxins fused with fluorescent proteins (FPs) [7,45,46,47]. Currently, FPs used for such constructions are limited to TagRFP (hereinafter RFP) [48] and enhanced GFP [49], while a list of peptide blockers includes MgTx, OSK1, hongotoxin 1, and AgTx2. Studies of these FP-tagged toxins (FP-Txs) reveal high (nanomolar and even subnanomolar) affinities of FP-Txs to particular isoforms of K_v_1 channels. It has been proved that FP-Tx binding to channels is accompanied by channel blocking. The relative affinities of FP-Txs to target channels may vary compared to those of a native toxin. Indeed, the affinities of FP-Txs to K_v_1 channels are lower than those of free toxins, but the extent of affinity reduction varies in a complicated way for particular combinations of FPs and toxins [7,45,46,47], for different FPs fused with the same toxin [7,46], and for different modes of tagging (N- or C-terminal) [46,47]. For example, N-terminal tagging of AgTx2 with RFP retained, in general, the relative affinities to the K_v_1.1, K_v_1.3, and K_v_1.6 channels but reduced moderately the relative affinity toward the K_v_1.2 channel [7]. In contrast, N-terminal tagging of AgTx2 with GFP resulted in considerable enhancement of the selectivity to the K_v_1.3 binding site compared to the K_v_1.1 and K_v_1.6 binding sites [46]. This ligand, GFP-AgTx2, is the first known example of an engineered ligand in which high selectivity for the target K_v_1 channel was achieved by simply fusing a peptide toxin with a bulky FP. It is noteworthy that the permutation of GFP from the N- to the C-terminus of AgTx2 restored interaction with the K_v_1.1 and K_v_1.6 binding sites [46].

These data clearly indicate that, when developing new FP-Txs based on various FPs and toxins, the spectrum of activities of new molecular constructions should be carefully studied to assess the effect of bulky FPs on the binding properties of the Tx moieties. FPs of different origins (for example, jellyfish or coral) have a specific surface charge and topology of charged residues that can affect the interactions of FP-Txs with channels.

Recently, we reported the design and preliminary properties of AgTx2 C-terminally fused with GFP (AgTx2-GFP) [46]. Using hybrid KcsA-K_v_1.x (x = 1, 3, 6) channels expressed in the inner membranes of *E. coli* cells, it was demonstrated that AgTx2-GFP interacted specifically with the pore blocker binding sites of K_v_1.x (x = 1, 3, 6) channels and was displaced from these complexes by known peptide blockers of these channels.

The present work is devoted to a detailed study of the properties of AgTx2-GFP. Affinities of AgTx2-GFP to hybrid KcsA-K_v_1.x (x = 1, 3, 6) channels on the membranes of *E. coli* spheroplasts, to mammalian K_v_1.x (x = 1–3, 6) channels on the membranes of *Xenopus laevis* oocytes, and to the human K_v_1.3 channel on the membranes of mammalian cells are evaluated. It is demonstrated that combinations of AgTx2-GFP and *E. coli* spheroplasts expressing KcsA-K_v_1.x (x = 1, 3, 6) channels or AgTx2-GFP and mammalian cells expressing the K_v_1.3 channel can be used for the search and study of peptide pore blockers. It is demonstrated that AgTx2-GFP is a high-affinity fluorescent probe for the imaging of K_v_1.3 channels in cells.

## 2. Results and Discussion

### 2.1. Interactions of AgTx2-GFP with KcsA-Kv1.x (x = 1, 3, 6) Channels

AgTx2-GFP was obtained by fusing GFP with the C-terminus of AgTx2 by means of a flexible 20 amino acid linker (GSGGSGGSGGTGGAGGATST) [46].

To characterize quantitatively the previously revealed interactions of AgTx2-GFP with the pore blocker binding sites of K_v_1.x (x = 1, 3, 6) channels, the concentration dependences of AgTx2-GFP binding to the hybrid potassium channels KcsA-K_v_1.x (x = 1, 3, 6) expressed in the inner membranes of *E. coli* cells were measured (Figure 1). KcsA-K_v_1.x (x = 1, 3, 6) channels comprised the pore blocker outer-binding sites of corresponding eukaryotic channels, which were transferred to the bacterial KcsA channel (Figure 1c) [44,50,51]. As shown earlier, the affinities of peptide pore blockers to KcsA-K_v_1.x (x = 1, 3, 6) channels were similar to their affinities to corresponding eukaryotic K_v_1 channels [52].

The confocal microscopy analysis of AgTx2-GFP interactions with KcsA-K_v_1.x (x = 1, 3, 6) channels on the membranes of spheroplasts prepared from *E. coli* cells revealed saturable binding that occurred in subnanomolar (KcsA-K_v_1.3) and low nanomolar (KcsA-K_v_1.1 and KcsA-K_v_1.6) concentration ranges (Figure 1d–f). This AgTx2-GFP binding was reversible, and peptide blockers of K_v_1.x (x = 1, 3, 6) channels displaced AgTx2-GFP from spheroplasts bearing KcsA-K_v_1.x (x = 1, 3, 6) channels [46]. In addition, AgTx2-GFP binding to the *E. coli* membrane itself, as well as to the KcsA channel embedded in the *E. coli* membrane, was negligible compared to its binding to spheroplasts bearing KcsA-K_v_1.x (x = 1, 3, 6) channels [46]. These findings allowed us to validly fit experimental data using Equation (1) (Figure 1d–f) and to determine the dissociation constants *K_d_* of AgTx2-GFP complexes with KcsA-K_v_1.x (x = 1, 3, 6) channels (Table 1).

The affinity of AgTx2-GFP to the hybrid channels reduced in the manner of KcsA-K_v_1.3 > KcsA-K_v_1.6 >> KcsA-K_v_1.1 and was found to be moderately sensitive to pH in the neutral to slightly basic pH range (Table 1). The affinity of AgTx2-GFP to KcsA-K_v_1.1 was the lowest at pH 7.0 and increased at higher pH values (Table 1). The affinities of AgTx2-GFP to KcsA-K_v_1.3 and KcsA-K_v_1.6 were highest at pH 7.5 and decreased at both pH 7.0 and pH 8.0 (Table 1). Since the *pK_a_* values of most amino acids are far from the pH range of 7–8 and cysteines of AgTx2 are in the form of cystines, the histidine residues with *pK_a_ =* 6.3 were the most likely candidates whose protonation–deprotonation equilibrium affected the *K_d_* of the studied complexes. H34 of AgTx2 and H58 of the binding site of KcsA-K_v_1.1 could be responsible for the revealed pH dependence, while the absence of histidine residues in the binding sites of KcsA-K_v_1.3 and KcsA-K_v_1.6 could be a reason for the variation in the character of pH dependences of these hybrid channels compared to KcsA-K_v_1.1 (Table 1). Additionally, a decrease in the protonation state of the N-terminal amino group of AgTx2, which occurred at the increase in pH from 7.0 to 8.0, could affect peptide binding.

As compared to previously studied FP-Txs, AgTx2-GFP was inferior to RFP-AgTx2 and GFP-OSK1 in its affinity to KcsA-K_v_1.1 (Figure 2a) [7]. At the same time, AgTx2-GFP surpassed all the studied FP-Txs in its reported affinities to KcsA-K_v_1.3 (Figure 2a) and KcsA-K_v_1.6 [7]. It should be noted that *K_d_* were measured for AgTx2-GFP at an ionic strength that was noticeably higher than in the measurements of *K_d_* values for other FP-Txs. A decrease in the ionic strength of the solution can increase the affinities of peptide blockers to the binding sites of K_v_1 channels due to the considerable role of electrostatic interactions in the formation of a peptide–channel complex [50,51].

AgTx2-GFP could be displaced from complexes with KcsA-K_v_1.x (x = 1, 3, 6) not only by pure, specific peptide pore blockers [46], but also by complex mixtures of natural peptides, such as crude scorpion venoms containing pore blockers (Figure 2b). This result confirms that AgTx2-GFP in combination with KcsA-K_v_1.x (x = 1, 3, 6) channels could be used to search for K_v_1 channel pore blockers among individual compounds and in complex natural extracts and venoms. As demonstrated earlier [51], the targeted detection of specific activity using a combination of a fluorescent ligand and KcsA-K_v_1.x (x = 1, 3, 6) channels facilitated significantly the targeted isolation of individual peptide blockers of K_v_1 channels from venoms.

Displacement of AgTx2-GFP from the complexes with KcsA-K_v_1.x (x = 1, 3, 6) channels by the nonlabeled peptide blocker occurred due to the competitive binding to the K_v_1 channel binding site. A quantitative analysis of such displacement by the studied peptide blocker for the particular channels (Figure 2c–e) allowed the estimation of the peptide concentration, which displaced 50% of AgTx2-GFP from the complexes (*DC_50_*), and the calculation of the apparent dissociation constant *K_ap_* of the studied peptide blocker (Equations (2) and (3)) [44]. The *K_ap_* values determined for KTx1 and AgTx2 using the AgTx2-GFP/KcsA-K_v_1.x (x = 1, 3, 6) bioengineering analytical systems (Table 2) allowed us to conclude that AgTx2-GFP was applicable as a component of these systems for the study of peptide blockers and their affinities to K_v_1 channels.

The affinities of AgTx2 to KcsA-K_v_1.3 and KcsA-K_v_1.6 estimated with AgTx2-GFP (Table 2) are in general agreement with the data on AgTx2 activity (*IC_50_,* blocker concentration inhibiting current through the channel by 50%) reported in electrophysiological studies: 4 pM [10], 50 pM [34], 200 pM [53], and 170 pM (Figure 3, Table 3) for K_v_1.3 and 37 pM [10] and 600 pM (Figure 3, Table 3) for K_v_1.6. The measured affinity of AgTx2 to KcsA-K_v_1.1 (Table 2) is close to the previously reported *K_ap_* value determined with another fluorescent ligand [51], as well as to recent *IC_50_* values of 2 nM [7] and 4.2 nM (Figure 3, Table 3) obtained in electrophysiological studies of the K_v_1.1 channel. At the same time, earlier electrophysiological studies have reported much lower *IC_50_* values (44 pM [10] and 130 pM [34]) for inhibition of the K_v_1.1 channel by AgTx2, and the reason for these differences is unclear.

The *K_ap_* values of KTx1 complexes with KcsA-K_v_1.1 and KcsA-K_v_1.3 (Table 2) correlated to the *IC_50_* values reported for KTx1 complexes with K_v_1.1 (1.1 nM) and K_v_1.3 (0.1 nM) [34]. To our best knowledge, there are no quantitative electrophysiological data on the inhibition of K_v_1.6 by KTx1, but only qualitative ones [54].

### 2.2. Electrophysiological Studies of AgTx2-GFP

According to the data on electrophysiological measurements on oocytes, AgTx2-GFP was a pore blocker of K_v_1.x (x = 1–3, 6) channels (Figure 3). Its affinity to the channels decreased in the following manner: K_v_1.6 > K_v_1.3 > K_v_1.1 >> K_v_1.2 (Table 3). C-terminal attachment of GFP did not affect the affinity of AgTx2 to K_v_1.1 but decreased its activity by two, ten, and more than thirty times to K_v_1.6, K_v_1.3, and K_v_1.2, respectively, thus changing noticeably the natural pharmacological profile of AgTx2 (Table 3). A significant decrease in the affinity of AgTx2-GFP to the Kv1.2 channel could be associated with the disturbance of strong interactions between the channel and amino acid residues near the C-terminus of the peptide by a volumetric fluorescent protein. According to 1H-NMR studies, these could be the Lys32 and His34 of AgTx2 forming a salt bridge with Asp355 and hydrophobic contacts with the Gly378, Asp379, and Val381 of the channel, respectively [55].

Finally, K_v_1.6, K_v_1.3, and K_v_.1.1 differed in affinity to AgTx2-GFP by less than three times, which made AgTx2-GFP a high-affinity fluorescent marker of Kv1 channels of a wide spectrum of action.

The affinities of AgTx2-GFP to hybrid KcsA-K_v_1.x (x = 1, 3, 6) channels (Table 1) and corresponding K_v_1.x (x = 1, 3, 6) channels (Table 3) differed noticeably, reflecting the influences of buffer composition and the structural differences of hybrid and natural channels on the binding of the bulky AgTx2-GFP ligand. Apparently, the FP (in our case, GFP) could participate in interactions with amino acid residues of the channel surrounding or beyond the natural binding site of the peptide blocker and, thus, modulate the overall affinity of the ligand to the channel. Differences in the amino acid compositions of KcsA-K_v_1.x and the corresponding K_v_1.x channels beyond the binding site led to differences in interactions with GFP moiety and, finally, to differences in the stability of the complexes.

### 2.3. Interaction of AgTx2-GFP with K_v_1.3 on Mammalian Cell Membranes

Recently, we created a plasmid encoding a natural (M1-D52)-truncated variant of the K_v_1.3 channel [56] fused with red fluorescent protein mKate2 (mKate2-K_v_1.3), which is effectively expressed in the plasma membranes of mammalian cells and retains voltage-gated K^+^ conductivity and the ability to bind peptide pore blockers [42]. Our studies show that AgTx2-GFP bound to the membrane of Neuro 2A and HEK293 cells expressing the mKate2-K_v_1.3 channel (Figure 4 and Appendix A). The K_v_1.3 channel was obviously a target of AgTx2-GFP in these cells since no binding of AgTx2-GFP to cells was observed in the absence of mKate2-K_v_1.3 channels (Appendix A). Moreover, AgTx2 displaced AgTx2-GFP from the surfaces of cells expressing the mKate2-K_v_1.3 channel, indicating competition of these ligands for the pore blocker binding site of K_v_1.3 (Figure 4 and Appendix A).

To characterize the affinity of AgTx2-GFP to K_v_1.3 on the membranes of mammalian cells, we measured the concentration dependence of AgTx2-GFP binding to mKate2-K_v_1.3-expressing Neuro 2A cells using confocal microscopy and calculated the ratios (*R_av_*) of the fluorescent intensity of bound AgTx2-GFP to the fluorescent intensity of membranous mKate2-K_v_1.3 at different concentrations of added AgTx2-GFP (Figure 5a). Previously, it was shown that the analysis of such a dependence using Equation (5) allowed one to calculate the *K_d_* of the formed complexes [42]. The measurements were carried out after a 30 min incubation of the cells with AgTx2-GFP when the complexation process reached equilibrium (Appendix A). The calculated *K_d_* of AgTx2-GFP complexes with K_v_1.3 on the Neuro 2A cell membranes was 3.4 ± 0.8 nM, which was consistent with the data of the electrophysiological measurements (Table 3).

It is worth mentioning that the measurements of the affinities of AgTx2-GFP to the K_v_1.3 channel with the fluorescent cell-based assay (Figure 5a) and electrophysiological approach (Figure 3) were performed at different states of the channel—closed and open, respectively. Similar values of *K_d_* and *IC_50_* obtained in these measurements allowed us to assume that the affinity of AgTx2-GFP to K_v_1.3 depended weakly, if at all, on the state of the channel (open or closed). To verify this assumption using the same technique, we compared changes in AgTx2-GFP binding to K_v_1.3 on the cell membranes in buffers containing either 150 mM NaCl or 150 mM KCl (Figure 5b). The presence of NaCl in the outer medium maintained the predominantly closed state of the channels, while KCl induced depolarization of the plasma membrane and the opening of the channels. The calculated *K_d_* values of the AgTx2-GFP complexes with open and closed K_v_1.3 channels were 3 ± 1 and 4.7 ± 0.8 nM, respectively. The difference between these *K_d_* values was insignificant, thus confirming the assumption made.

Considering a probable application of AgTx2-GFP as a fluorescent ligand for the study of interactions of various peptide blockers with K_v_1.3 channels on mammalian cells, we measured the concentration dependence of the competitive binding of AgTx2 to K_v_1.3 at a constant concentration (8 nM) of AgTx2-GFP added to cells following an earlier developed procedure [42]. An analysis of the displacement of AgTx2-GFP from complexes with mKate2-K_v_1.3 by increasing concentrations of AgTx2 allowed one to calculate the AgTx2 concentration displacing 50% of the AgTx2-GFP from the complexes (*DC_50_*, Equation (6)) and the *K_ap_* of the complexes between AgTx2 and K_v_1.3 (Equation (7)). The calculated *K_ap_* was 0.7 ± 0.2 nM, which is in a good agreement with the electrophysiological data presented in Table 3 and published earlier [10,34,53].

The data obtained allowed one to conclude that AgTx2-GFP was suitable for the fluorescent imaging of K_v_1.3 channels in mammalian cells, as well as for recognizing K_v_1.3 blockers among the studied peptide toxins and assessing their affinity for K_v_1.3, when AgTx2-GFP was used in combination with mammalian cells expressing fluorescent (FP-tagged) K_v_1.3 channels. Such studies are also useful at an earlier stage of the selection of a candidate drug among peptides. In vitro cell experiments are a mandatory part of preclinical studies for any candidate drug. The initial selection and study of peptide blockers is often performed in buffer solutions by means of electrophysiological techniques on *Xenopus* oocytes or radioligand analyses on isolated membranes. These results need to be verified in a more complex and natural environment, where background proteolytic activity and various nonspecific interactions with the components of a mammalian cell growth medium can affect the blocker–channel interactions.

AgTx2-GFP is a high-affinity fluorescent ligand of the K_v_1.1, K_v_1.3, and K_v_1.6 channels whose increased expression and activity are associated with a number of diseases. The K_v_1.3 channel is involved in the development of autoimmune or neuroimmune inflammation through the induction of pro-inflammatory response and the activation of T lymphocytes or microglial cells, respectively [57,58]. CD4+ effector memory T cells mediate many autoimmune diseases, including multiple sclerosis, type-I diabetes mellitus, and rheumatoid arthritis. The selective blockage of K_v_1.3 channels in these cells inhibits their proliferation and cytokine production, thereby contributing to a reduction in inflammation and providing a therapeutic effect [23]. The pathogeneses of oncological diseases, such as breast and prostate cancers, pancreatic ductal adenocarcinoma, or B-type chronic lymphocytic leukemia, are associated with the upregulation of K_v_1.3 channel expression [59]. The inhibition of K_v_1.3 channels by either small organic blockers or peptide toxins from animal venoms results in a significant therapeutic effect without major side effects [60]. Inherited loss-of-function mutations of the K_v_1.1 channel in humans are associated with episodic ataxia type I [61], while some other deletions or mutations of the K_v_1.1 and K_v_1.2 channels provoke epilepsy [62]. Although the blockage of K_v_1.1 and K_v_1.2 channels has no therapeutic value in cases of increased excitability of neurons, the inhibition of K_v_1.1 channels in inhibitory (demyelinated) axons is known to improve neuronal conduction in multiple sclerosis [63]. While the physiological role of the K_v_1.6 channel, which is widely expressed in the central nervous system and peripheral neurons, is still poorly determined, the localization of K_v_1.6 in microglial cells in the striatum suggests its potential role in neuroinflammation and striatal disorders, such as Parkinson’s disease [64].

Comparing the prospects of the best organic fluorophores and FPs for the fluorescent labeling of peptide ligands of K_v_1 channels, it can be stated that organic fluorophores are usually brighter and more photostable, but the differences are gradually being leveled due to the extensive development of new FP variants [65,66,67]. For many laboratories that are familiar with the technology of recombinant proteins, FP-Tx production is simpler than the synthesis of fluorophore-labeled peptide blockers, especially because the recombinant production of FP-Txs usually provides a high yield of a fully active ligands with properly formed disulfide bonds without additional refolding of the product [46]. It should be noted that even small organic fluorophores can unpredictably modulate the affinities and pharmacological profiles of peptide blockers of K_v_1 channels [43], not only FPs [46].

## 3. Conclusions

AgTx2-GFP was shown to have subnanomolar affinities to hybrid KcsA-K_v_1.3 and KcsA-K_v_1.6 channels and a nanomolar affinity to the KcsA-K_v_1.1 channel (Table 1). Since these affinities were slightly sensitive to pH in the 7.0–8.0 range, a constant pH value should be maintained in studies to increase the reproducibility of the results. AgTx2-GFP could be used as a component of the analytical system based on hybrid Kcsa-K_v_1.x (x = 1, 3, 6) channels expressed in the membranes of *E.coli* cells [52] for the study of peptide blockers and the evaluation of their affinities to K_v_1 channels (Table 2). Data from electrophysiological measurements on oocytes indicated that AgTx2-GFP was a pore blocker of K_v_1.x (x = 1, 3, 6) channels possessing nanomolar activity (Table 3). AgTx2-GFP could be equally used as a high-affinity fluorescent marker of the K_v_1.6, K_v_1.3, and K_v_.1.1 channels. Nanomolar activity of AgTx2-GFP was preserved for both open and closed K_v_1.3 channels expressed in the membranes of mammalian cells (Figure 5). Fluorescent imaging of K_v_1.3 channels and studies of the activities of K_v_1.3 peptide blockers are potential applications of AgTx2-GFP in mammalian cells.

Data on the properties of AgTx2-GFP and previously studied FP-Txs [7,45,46,47] have shown that FP-Txs can be considered as a reliable alternative to peptide blockers labeled with organic fluorophores for the fluorescent imaging of K_v_1 channels, as well as for a number of analytical applications. FPs can modulate the pharmacological profiles of peptide blockers of K_v_1 channels, maintaining a high (nanomolar) affinity to some of the target channels or even enhancing considerably the ligand selectivity for a particular channel. A wide variety of peptide blockers differing in the repertoire of amino acid residues interacting with K_v_1 channels, together with a wide variety of FPs having not only different colors but also distinct compositions and distributions of surface-charged residues, can provide multiple combinations of FP-Txs with valuable properties. The design and study of such FP-Txs is still at an early stage.

## 4. Materials and Methods

### 4.1. Reagents

AgTx2-GFP was produced as described previously [46]. Its concentration in an aqueous solution was determined using the molar extinction coefficient of 55,000 M^−1^cm^−1^ at 489 nm. Recombinant peptides of KTx1 and AgTx2 were obtained as described earlier [68]. Concentrations of KTx1 and AgTx2 were measured in an aqueous solution containing 20% acetonitrile and 0.1% trifluoroacetic acid using molar extinction coefficients (at 214 nm) of 49,300 and 49,200 M^−1^ cm^−1^, respectively.

The plasmid pmKate2-KCNA3-del was obtained as described earlier [42].

Crude venoms of *M. eupeus* and *O. scrobiculosus* scorpions were a kind gift from Dr. Alexander Vassilevski.

### 4.2. Experiments with KcsA and KcsA-K_v_1.x (x = 1, 3, 6) Channels

Hybrid KcsA-K_v_1.x (x = 1,3,6) channels were expressed in *E. coli* cells as described earlier [44,50,51]. Cells were transformed to spheroplasts following the protocol described earlier [44]. For AgTx2-GFP binding experiments, spheroplasts expressing KcsA-K_v_1.x (x = 1, 3, 6) channels were incubated (500–1000 cells/μL, 37 °C, 2 h) with AgTx2-GFP (0.02–85 nM) in a buffer containing 50 mM NaCl, 50 mM Tris-HCl (pH 7.0, 7.5, or 8.0), 0.1% BSA, 0.25 M sucrose, 10 mM MgCl_2_, 4 mM KCl, and 0.3 mM EDTA. Displacement of AgTx2-GFP from complexes with KcsA-K_v_1.x (x = 1, 3, 6) channels was analyzed by incubating spheroplasts with a constant concentration of AgTx2-GFP and increasing concentrations of AgTx2 or KTx1 peptides (0.04–100 nM) at 37 °C for 2 h. In similar experiments with scorpion venoms, AgTx2-GFP was added to spheroplasts, together with crude venom (40 mg/L) of *M. eupeus* or *O. scrobiculosus* scorpions. Concentrations of AgTx2-GFP in the experiments with KcsA-K_v_1.1, KcsA-K_v_1.3, and KcsA-K_v_1.6 were 80, 8, and 2 nM, respectively. Measurements were performed at pH 7.5 in the buffer described above.

Spheroplasts were imaged with a laser scanning confocal microscope LSM710 (Zeiss, Germany) equipped with an α Plan-Apochromat 100×/1.46 oil immersion objective (excitation at 488 nm, detection in the 495–590 nm range, 0.2 µm lateral and 2 µm axial resolutions). Fluorescent images of spheroplasts were analyzed as reported earlier [44,50,51]. Fluorescence intensity of AgTx2-GFP bound to KcsA-K_v_1.x (x = 1, 3, 6) was estimated for each analyzed spheroplast and averaged over 150–250 spheroplasts to obtain the *I_a_* value (±SD).

*K_d_* values of AgTx2-GFP complexes with KcsA-K_v_1.x (x = 1, 3, 6) were calculated from the measured *I_a_* dependencies on the concentration *L* of added AgTx2-GFP using the following equation:I_a_ = I_s_ L/(K_d_ + L) (1)
where *I_s_* is the *I_a_* value at the saturation of binding.

Data on the displacement of AgTx2-GFP from complexes with KcsA-K_v_1.x (x = 1, 3, 6) by a nonlabeled peptide blocker were fitted with the following equation:*I_a_* = *I_0_*/(1 + *B*/*DC_50_*) (2)
where *B* is the concentration of the added nonlabeled peptide, *DC_50_* is the concentration of the nonlabeled peptide displacing 50% of AgTx2-GFP from the complex with the KcsA-K_v_1.x channel, and *I_0_* is the *I_a_* at *B* = 0.

The *K_ap_* value of the nonlabeled peptide blocker was calculated using the Cheng–Prusoff equation:*K_ap_* = *DC_50_*/(1 + *L*/*K_d_*)(3)
where *L* is the constant concentration of AgTx2-GFP.

The *K_d_* and *K_ap_* values were averaged over three independent experiments and presented as means ± SEM.

### 4.3. Electrophysiological Experiments

Stage V–VI oocytes were harvested from an anesthetized female *X. laevis* frog following the KU Leuven guidelines for animal welfare, as described earlier [69,70]. The oocytes were incubated in ND96 solution (96 mM NaCl, 2 mM KCl, 1.8 mM CaCl_2_, 2 mM MgCl_2_, 5 mM HEPES, pH 7.4) containing 50 mg/L gentamycin sulfate.

Expressions of rat K_v_1.1, K_v_1.2, and K_v_1.6 and human K_v_1.3 channels in *X. Laevis* oocytes were performed as described previously [71]. Transcription of linearized plasmids bearing K_v_1 genes was performed using either T7 or SP6 mMESSAGE mMACHINE transcription kits (Invitrogen, CA, USA).

Two-electrode voltage clamp recording was performed at 18–22 °C using a Geneclamp 500 amplifier (Molecular Devices, CA, USA) and an Axon pClamp data acquisition system (Molecular Devices, CA, USA), as described earlier [72]. Whole-cell currents from oocytes were recorded 1–4 days after injection of 50 nl of cRNA (1 g/L). Current and voltage electrodes were filled with 3 M KCl, and their resistances were kept between 0.7 and 1.5 MΩ. The bath solution was ND96. The induced currents were filtered at 0.5 kHz and sampled at 2 kHz using a four-pole low-pass Bessel filter. Leak was subtracted using a—P/4 protocol. Currents through K_v_1.x (x = 1–3, 6) channels were induced by 250 ms depolarization to 0 mV followed by a 250 ms pulse to −50 mV from a holding potential of −90 mV.

Concentration dependences of the inhibition of currents through K_v_1.x (x = 1–3, 6) channels by AgTx2-GFP and AgTx2 were measured and fitted with the Hill equation:*y* = 100/(1 + (*IC_50_*/*C*)*^h^*) (4)
where *y* is the relative decrease (%) in current amplitude at the concentration *C* of a pore blocker, *IC_50_* is the pore blocker concentration inducing a 50% decrease in the current amplitude, and *h* is the Hill coefficient. The Hill coefficients for all the measured dependences were found to be close to 1 (not shown). Concentrations of AgTx2-GFP and AgTx2 varied from 1 nM to 1 µM. Experimental data obtained in several independent experiments are presented as means ± SEM (n ≥ 3).

### 4.4. Experiments with Neuro2a Cells Expressing K_v_1.3 Channels

Mouse Neuro-2A cells (from the collection of the Institute of Cytology RAS, Russia) were grown (37 °C, 5% CO_2_) in Dulbecco’s modified Eagle’s medium DMEM/F12 (Paneco, Russia) supplemented with 5% fetal bovine serum (FBS, HyClone, UT, USA) and 2 mM L-glutamine (complete medium). Cells were replanted every 72 h. Cells used in the experiments were from the 5–20 passages. Transient transfection with the pmKate2-KCNA3-del plasmid was performed using GenJector-U reagent (Molecta, Russia) at nearly 50% confluence according to the manufacturer’s protocol. All the experiments with cells were performed within 24 h after transfection. Cells were incubated with AgTx2-GFP (1–40 nM) in complete medium at 37 °C for 30 min. In the competitive binding experiments, cells were incubated with AgTx2-GFP (8 nM) and different concentrations (0.5–40 nM) of AgTx2 in complete medium at 37 °C for 30 min. In the experiments on the influence of channel state (open vs. closed) on ligand binding, cells were incubated with AgTx2-GFP (2–40 nM, 37 °C, 30 min) in a buffer (20 mM HEPES pH 7.4, 0.9 mM CaCl_2_, 0.5 mM MgCl_2_, and 0.1% BSA) containing either 150 mM KCl or 150 mM NaCl and 4 mM KCl.

Fluorescent images of cells were recorded with a laser scanning confocal microscope Leica-SP2 (Leica Microsystems GmbH, Wetzlar, Germany) equipped with a water immersion 63×/1.2 NA HCX PL APO objective (0.2 µm lateral and 0.6 µm axial resolutions). Fluorescence of AgTx2-GFP was excited at the 488 nm wavelength and recorded in the 495–535 nm range. Fluorescence of mKate2-K_v_1.3 was excited at the 561 nm wavelength and recorded in the 650–700 nm range. Sequential scanning was used.

Fluorescent images were treated using ImageJ software (National Institutes of Health, Bethesda, MD, USA). Average fluorescence intensities of membrane-bound AgTx2-GFP and mKate2-K_v_1.3 were determined for each examined cell and corrected for background signal, as described earlier [42], and their ratios *R_i_* were calculated. The *R_i_* values of 20–25 cells measured in the same conditions were averaged, producing the *R_av_* values (±SD). The *R_av_* values were plotted as a function of the concentration *L* of added AgTx2-GFP (Figure 5a) and fitted with the following equation:*R_av_* = *R_m_L*/(*K_d_* + *L*) (5)
where *R_m_* is *R_av_* at the saturation of AgTx2-GFP binding.

The *R_av_* values were plotted as a function of the concentration *B* of added AgTx2 at a fixed concentration *L* of AgTx2-GFP (Figure 5c) and fitted with the following equation:R_av_ = R_av0_/(1 + B/DC_50_) (6)
where *DC_50_* is the concentration of AgTx2 displacing 50% of AgTx2-GFP from the complex with the K_v_1.3 channel, and *R_av0_* is *R_av_* at *B* = 0.

The *K_ap_* value of AgTx2 was calculated using the Cheng–Prusoff equation:K_ap_ = DC_50_/(1 + L/K_d_)(7)

All the measurements were repeated in two independent experiments, and data on the *K_d_* and *K_av_* were averaged and presented as means ± SEM.

## Figures and Tables

**Figure 1 toxins-15-00229-f001:**
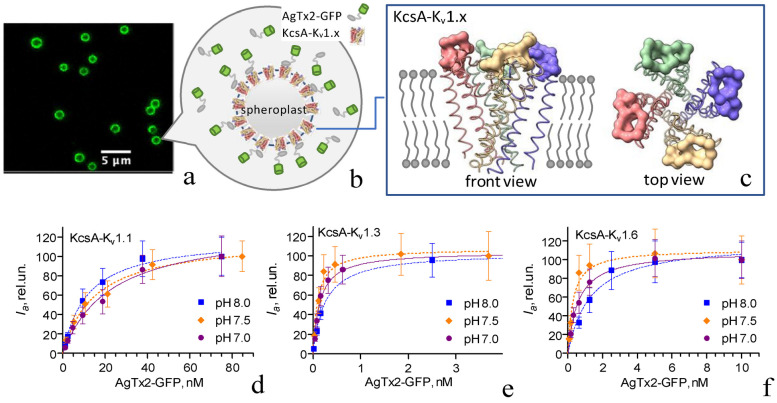
Interactions of AgTx2-GFP with hybrid KcsA-K_v_1.x (x = 1, 3, 6) channels on the membranes of *E. coli* spheroplasts. (**a**) Typical confocal image of AgTx2-GFP bound to KcsA-K_v_1.x on the membranes of *E. coli* spheroplasts. (**b**) A scheme of AgTx2-GFP interaction with a spheroplast expressing KcsA-K_v_1.x channels on the plasma membrane. (**c**) A scheme of the structure of a hybrid KcsA-K_v_1.x channel (based on PDB:1F6G of the KcsA channel). Subunits of the KcsA-K_v_1.x channel are shown in different colors. KcsA residues are shown in ribbon presentation. Positions of K_v_1.x residues transferred to KcsA are shown in cloud presentation. (**d**–**f**) Concentration dependences of AgTx2-GFP binding to hybrid KcsA-K_v_1.x (x = 1, 3, 6) channels at pH 7.0, 7.5, and 8.0. *I_a_* values are presented as means ± SD.

**Figure 2 toxins-15-00229-f002:**
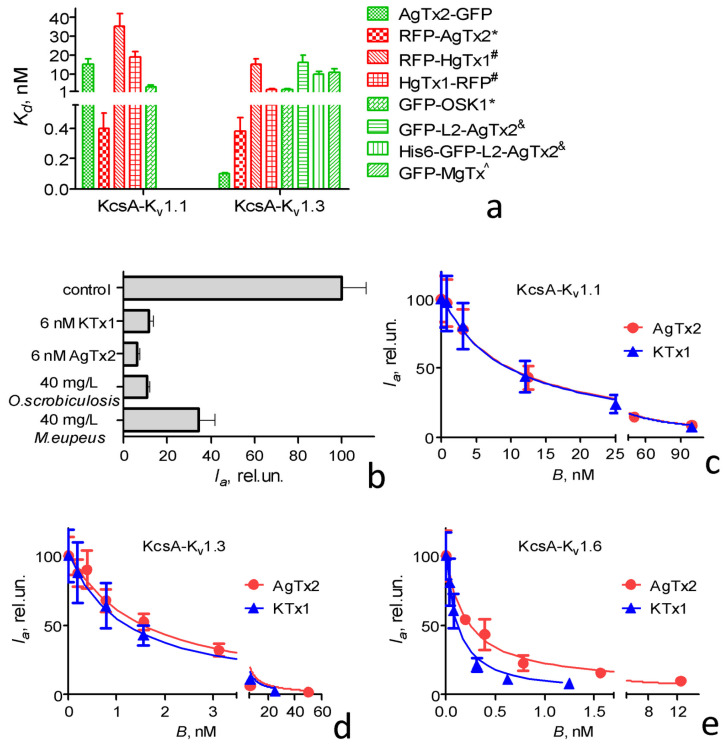
AgTx2-GFP as a ligand of KcsA-K_v_1.x (x = 1, 3, 6) channels. (**a**) Comparison of dissociation constants (*K_d_*, mean ± SEM) of KcsA-K_v_1.x (x = 1, 3) complexes with AgTx2-GFP and with previously studied FP-tagged pore blockers. * [7], ^#^ [47], ^&^ [46], ^^^ [45]. (**b**) Displacement of AgTx2-GFP (5 nM) from complexes with KcsA-K_v_1.3 (control) by kaliotoxin 1 (KTx1), AgTx2, and the components of crude venoms of *Mesobuthus eupeus* and *Orthochirus scrobiculosus* scorpions. (**c**–**e**) Displacement of AgTx2-GFP from complexes with KcsA-K_v_1.1 (**c**), KcsA-K_v_1.3 (**d**), and KcsA-K_v_1.6 (**e**) by different concentrations *B* of KTx1 and AgTx2. *I_a_*—relative fluorescence intensity of AgTx2-GFP bound to spheroplasts bearing KcsA-K_v_1.x (x = 1, 3, 6) channels. Concentrations of AgTx2-GFP in the experiments with KcsA-K_v_1.1, KcsA-K_v_1.3, and KcsA-K_v_1.6 were 80, 8, and 2 nM, respectively. *I_a_* values are presented as means ± SD.

**Figure 3 toxins-15-00229-f003:**
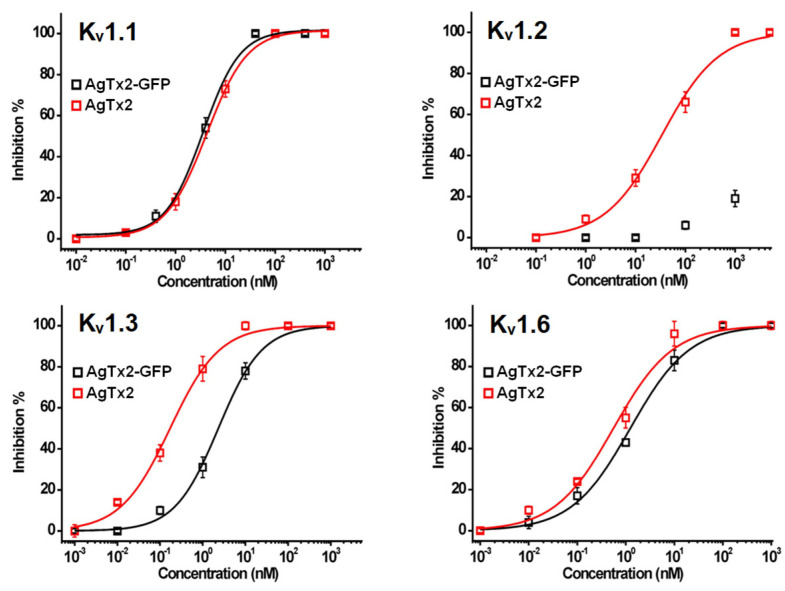
Concentration dependences of current inhibition through K_v_1.x (x = 1–3, 6) channels in oocytes by AgTx2 and AgTx2-GFP measured with a two-electrode voltage clamp technique. Currents through K_v_1.x (x = 1–3, 6) channels were induced by 250 ms depolarization to 0 mV followed by a 250 ms pulse to −50 mV from a holding potential of −90 mV. Data are presented as means ± SEM and fitted with Equation (4) (see Materials and Methods).

**Figure 4 toxins-15-00229-f004:**
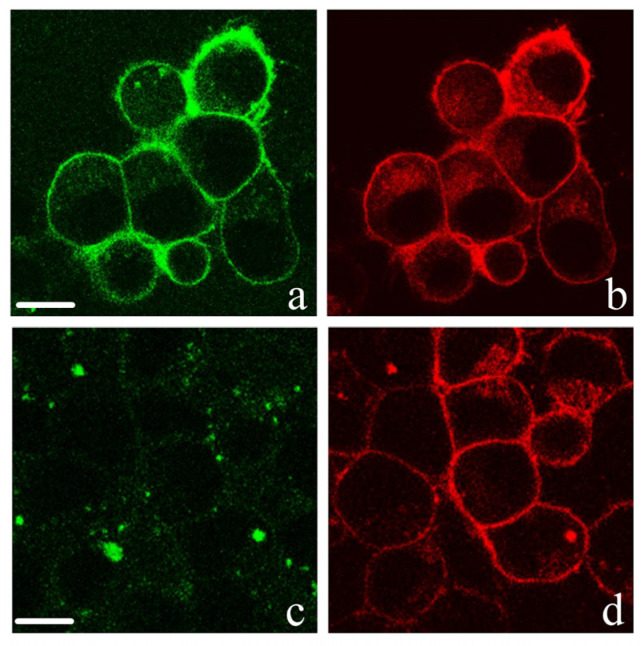
Confocal imaging of the interaction of AgTx2-GFP (green) with Neuro 2A cells expressing mKate2-K_v_1.3 (red). (**a**,**b**) Distribution of AgTx2-GFP (8 nM, 30 min incubation) bound to K_v_1.3 on the membranes of cells. Bar is 10 µm. (**c**,**d**) Concurrent inhibition of the binding of AgTx2-GFP (8 nM) to K_v_1.3 by AgTx2 (8 nM). Incubation time of cells with the ligands was 30 min. Bar is 10 µm.

**Figure 5 toxins-15-00229-f005:**
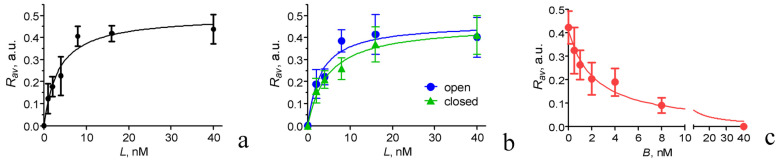
Quantitative analysis of AgTx2-GFP interaction with the Kv1.3 channel on the membranes of Neuro 2A cells expressing mKate2-K_v_1.3 channels. (**a**) Concentration dependence of AgTx2-GFP binding to K_v_1.3 measured in a culture medium as the dependence of *R_av_* (mean ± SD) on the concentration *L* of added AgTx2-GFP (see Section 4 for details). (**b**) Comparison of AgTx2-GFP binding to K_v_1.3 (*R_av_*, mean ± SEM) on the membranes of Neuro 2A cells expressing mKate2-K_v_1.3 channels in conditions that favored either open or closed states of the channels. Measurements were performed in a buffer (20 mM HEPES pH 7.4, 0.9 mM CaCl_2_, 0.5 mM MgCl_2_, and 0.1% bovine serum albumin) containing either 150 mM KCl or 150 mM NaCl and 4 mM KCl, thus either favoring open or maintaining closed states of the channels, respectively. (**c**) Concentration dependence of the displacement of AgTx2-GFP (8 nM) from complexes with K_v_1.3 channels by AgTx2 measured as the dependence of *R_av_* (mean ± SD) on the concentration *B* of added AgTx2 (see Section 4 for details).

**Table 1 toxins-15-00229-t001:** Dissociation constants *K_d_* (nM) of AgTx2-GFP complexes with KcsA-K_v_1.x (x = 1, 3, 6) channels at different pH values.

pH	KcsA-K_v_1.1	KcsA-K_v_1.3	KcsA-K_v_1.6
7.0	23 ± 4	0.14 ± 0.02	0.60 ± 0.06
7.5	15 ± 3 *	0.10 ± 0.01 *	0.30 ± 0.08 *
8.0	12 ± 2 *	0.25 ± 0.02 *	1.3 ± 0.4 *

* *p* < 0.05, comparison with *K_d_* at pH 7.0.

**Table 2 toxins-15-00229-t002:** Apparent dissociation constants *K_ap_* (nM) of AgTx2 and KTx1 complexes with KcsA-K_v_1.x (x = 1, 3, 6) channels.

Toxin	KcsA-K_v_1.1	KcsA-K_v_1.3	KcsA-K_v_1.6
AgTx2	3.7 ± 0.8	0.039 ± 0.007	0.027 ± 0.004
KTx1	3.8 ± 0.8	0.032 ± 0.006	0.016 ± 0.002

**Table 3 toxins-15-00229-t003:** Concentrations of AgTx2 and AgTx2-GFP that induced 50% inhibition of the current through K_v_1.x (x = 1–3,6) channels in oocytes (*IC_50_*, nM).

Ligand	K_v_1.1	K_v_1.2	K_v_1.3	K_v_1.6
AgTx2-GFP	3.5 ± 0.5 ^1^	- ^2^	2.31 ± 0.22	1.30 ± 0.15
AgTx2	4.2 ± 0.3	34 ± 6	0.17 ± 0.03	0.60 ± 0.11

^1^ mean ± SEM. ^2^
*IC_50_* was not achieved. 1 µM AgTx2-GFP provided 19% inhibition.

## Data Availability

The data presented in this study are available on request from the corresponding author. The data are not publicly available due to local regulations.

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
