# Peer review of "AgTx2-GFP, Fluorescent Blocker Targeting Pharmacologically Important Kv1.x (x = 1, 3, 6) Channels"

_toxins, 2023, doi:10.3390/toxins15030229_

Round 1

Reviewer 1 Report

In this manuscript, authors have developed a molecular tool based on Agitoxin to modulate potassium channels Kv1.x family. They have generated a fluorescent protein tagged Agitoxin (AgTx2-GFP) which can be genetically encoded as well as purified for biochemical and invitro applications. In summary, this is a timely and interesting piece of work. I have a few suggestions for the authors.

1.       Can authors speculate why the AgTx2-GFP performs so badly in inhibiting Kv1.2 compared to the untagged AgTx2 (Figure 3) ?

2.       It will be good to add a time dependence of AgTx2-GFP loading and displacement (upon adding AgTx2) in the imaging data (Figure 4).

3.       Are the images in Figure 4 top vs bottom panels acquired using different magnifications, the scale bars are different sized. Please correct this or clarify in captions.

Author Response

  1. Can authors speculate why the AgTx2-GFP performs so badly in inhibiting Kv1.2 compared to the untagged AgTx2 (Figure 3) ?

A significant decrease in the affinity of AgTx2-GFP to the Kv1.2 channel may be associated with a disturbance of strong interactions between the channel and amino acid residues near the C-terminus of the peptide by a volumetric fluorescent protein. According to 1H-NMR studies, these may be Lys32 and His34 forming a salt bridge with Asp355 and hydrophobic contacts with Gly378, Asp379 and Val381 of the channel, respectively [Pimentel  at al. Protein Sci. 2008;17(1):107-18. doi: 10.1110/ps.073122908].

  1. It will be good to add a time dependence of AgTx2-GFP loading and displacement (upon adding AgTx2) in the imaging data (Figure 4).

Legend to Figure 4 was slightly modified. Also, we have evaluated a time dependence of AgTx2-GFP binding to Kv1.3 channels and added these data in Supplementary section as Figure S3. Complexation process reaches equilibrium for less than 15 min. In our study, all the measurements were carried out after 30 min incubation of cells with AgTx2-GFP.  As for the time dependence of the AgTx2-GFP displacement from the complexes with Kv1.3 by AgTx2, please, note that we have added AgTx2-GFP and AgTx2 to cells simultaneously in our experiments. In these conditions, competitive binding occurs and not the displacement of AgTx2-GFP by AgTx2.In our preliminary experiments we determined that 30 min incubation is sufficient to achieve equilibrium binding in such reaction mixture.  

  1. Are the images in Figure 4 top vs bottom panels acquired using different magnifications, the scale bars are different sized. Please correct this or clarify in captions.

The images were acquired in the same conditions. The sizes of cropped areas were different. Now they are equal.   

Reviewer 2 Report

The manuscript presented is an interesting work that deals with the study of the binding of AgTx2-GFP (a fluorescent toxin blocker) to Kv potassium channels, showing its validity to perform studies on visualization of these proteins, or on searching non-labelled pore blockers of these channels, including measurements of their affinities..

The techniques used are suited for the study, controls are properly done in general, and conclusions are consistent with the results obtained. There are just minor questions that authors should address before considering its publication.

-        Table 1 is a summary of binding results at different pHs. Why did you use these experiments between 7 and 8.5? why these pHs?

-        Line 155-162, the authors claim that ionic strength can influence the affinity of the peptide blockers to the Kv pore. Have you performed measurements of binding affinity at different ionic strengths?

-        In figures 1, 2, 3, explain error bars. Are the measurements the average +- SD? SEM?

-        Figure 3 footnote is quite poor and more information is needed. Explain also the Fits?

-        Table 3, for Kv1.6 error with just a significant number.

-        Line 277, you mention the use of NaCl vs KCl in the extracellular medium as a way to open or close the channel. Why do not show you a complete curve of Rav vs L in NaCl vs KCl and not just a single point?

-        Line 458, “Kd and Kav were averaged”, and data showed +- sem?

Author Response

-        Table 1 is a summary of binding results at different pHs. Why did you use these experiments between 7 and 8.5? why these pHs?

Usually buffers that are used for in vitro experiments by various scientific groups have pH in the range from 7.0 to 7.8. Trying to improve the accuracy of the measured dissociation constants, we found that the results are sensitive to pH changes even in the 7.0-8.0 range (Table 1). It cannot be excluded that measurements by other methods (electrophysiological, radioligand analysis) also require more precise control of the pH value.

-        Line 155-162, the authors claim that ionic strength can influence the affinity of the peptide blockers to the Kv pore. Have you performed measurements of binding affinity at different ionic strengths?

Yes, we did it. See, for example, Figs. 2, 3 and corresponding text in Nekrasova et al. 2017 (DOI 10.1007/s11481-016-9710-9) as well as Fig.2 and corresponding text in Kuzmenkov et al. 2015 (DOI 10.1074/jbc.M115.637611).

-        In figures 1, 2, 3, explain error bars. Are the measurements the average +- SD? SEM?

Done.

-        Figure 3 footnote is quite poor and more information is needed. Explain also the Fits?

Figure legend was corrected.

-        Table 3, for Kv1.6 error with just a significant number.

Here we used a rule: the error of the measurement result is indicated by two significant digits if the first of them is 1 or 2, and one if the first digit is 3 or more. In the first case a rule used for rounding of an average value is: 4.21±0.12 is correct, 4.2±0.12 or 4.2±0.1 are not correct.   

-        Line 277, you mention the use of NaCl vs KCl in the extracellular medium as a way to open or close the channel. Why do not show you a complete curve of Rav vs L in NaCl vs KCl and not just a single point?

The corresponding curves were measured and presented in Figure 5 b. Dissociation constants were calculated and found to be insignificantly different.  

-        Line 458, “Kd and Kav were averaged”, and data showed +- sem?

Exactly. Now it is indicated.  

Reviewer 3 Report

see the attachment

Author Response

ï‚· Add more examples of potent blockers in respective portion.

Done.

ï‚· Revise the conclusion section of this article.

Done.

ï‚· Where you have mentioned equations, please write in detail about coefficients.

We checked the descriptions of the equations again. Each symbol is described and explained.

ï‚· Recheck all references, as the format of references is not uniform.

We do it every time. But Mendeley plugin for references correct them back time to time.  In any case, we will finally solve the problem at the latest stage. 

ï‚· Extensive English improvement is needed.

We did our best to improve it according to your comments and suggestions. 

The following are some specific comments in details:

  1. In line 28, Introduction section needs to be revised as there is no connectivity in some text portions.

Revision of Introduction section was performed in accordance with the reviewer’s comments and suggestions.

  1. In line 31, “The toxins bind at the external vestibule of the channel pore, physically occluding the pore.” needs to be rephrased.

Corrected as: “The toxins bind at the outer vestibule of a channel, occluding the pore with a lysine residue.”

  1. In line 35, “that scorpion toxins are able to discriminate a few targets from dozens of different K+ channels” this point regarding scorpion toxins ability to discriminate particular potassium channels needs more explanation.

Two typical examples of such toxins are now presented. 

  1. In line 42, enlist few more potent blockers for different relevant channels.

We did it by listing additionally, iberiotoxin, scyllatoxin, Vm24 and HsTX1[R14A], and giving examples of their applications. 

  1. In lines 46-49, “AgTx2 together with margatoxin helped to clarify a role of Kv1.1 and Kv1.3 channels in the mechanisms of retinal ganglion degeneration [16] and to determine functional expression of Kv1.3 by activated hippocampal microglia in the experimental model of epilepsy” rephrase these lines as the meaning is not clear.

Rephrased as:

AgTx2 and margatoxin helped to prove involvement of Kv1.1 and Kv1.3 channels in the mechanisms of retinal ganglion cell degeneration [18] and to identify expression of Kv1.3 in activated microglial cells as a probable reason of modulation of microglia proliferation [19].

  1. In lines 53-55, “it was shown that therapy-oriented selective peptide blockers can be designed combining a scaffold-based strategy and randomized mutagenesis on the platform of the phage display peptide-based library” needs rephrasing.

Corrected as

It has been shown that therapy-oriented selective peptide blockers can be developed using phage display based on the library of scorpion toxin sequences.  

  1. In line 59, the word “trafficking” should be rechecked for proper meaning here.

Changed to “traffic”

  1. In line 73, the word “repertoire” should be replaced with some other suitable word.

Changed to “a list”

  1. From lines 74-77, the sentence “Studies of these FP-tagged toxins (FP-Txs) revealed high (nanomolar and even subnanomolar) affinity of FP-Txs to particular isoforms of Kv1 channels and preservation of channel blocking activity, which may be accompanied by changes in a pharmacological profile as compared to free toxins” should be rephrased for clear understanding.

Corrected as

Studies of these FP-tagged toxins (FP-Txs) revealed high (nanomolar and even subnanomolar) affinity of FP-Txs to particular isoforms of Kv1 channels. It has been proven that FP-Tx binding to channels is accompanied by channel blocking. The relative affinity of FP-Tx to target channels may vary compared to those of a native toxin.

  1. Lines 90-93 should be rephrased because the meaning is not clear yet.

Corrected as

These data clearly indicate that when developing new FP-Txs based on various fluorescent proteins and toxins, the spectrum of activities of new molecular constructions should be carefully studied to assess the effect of bulky FP on the binding properties of the Tx moiety.  FPs of different origin (for example, jellyfish, coral) have a specific surface charge and topology of charged residues, that can affect interactions of FP-Txs with channels.

  1. From lines 100-105, the last paragraph of ‘introduction’ should be revised for clear meaning.

Revised as

The present work is devoted to the detailed study of the properties of AgTx2-GFP. Affinities of AgTx2-GFP to hybrid KcsA-Kv1.x (x=1,3,6) channels on the membrane of E. coli spheroplasts, to mammalian Kv1.x (x=1-3,6) channels on the membrane of Xenopus laevis oocytes and to human Kv1.3 channel on the membrane of mammalian cells are evaluated. It is demonstrated that combination of AgTx2-GFP and E. coli spheroplasts expressing KcsA-Kv1.x (x=1,3,6) channels, or AgTx2-GFP and mammalian cells expressing Kv1.3 channel can be used for the search and study of peptide pore blockers. It is demonstrated that AgTx2-GFP is a high affinity fluorescent probe for imaging of Kv1.3 channels in cells. 

  1. In line 211, “channels decreases in the row” should be decrease in a manner.

Corrected.

  1. From line 275-277, the sentence “To verify this assumption in similar experimental conditions, we have compared changes in the AgTx2-GFP binding to Kv1.3 on the cell membrane at150 mM concentration of NaCl and KCl in the extracellular medium” should be written in a concise way to understand clearly.

Corrected as

To verify this assumption using the same technique, we have compared changes in the AgTx2-GFP binding to Kv1.3 on the cell membrane in the buffer containing either 150 mM NaCl or 150 mM KCl.

  1. The statement in lines 282-285 should be explained more for understanding.

This text was completely changed, since concentration dependences of AgTx2-GFP binding to open and closed Kv1.3 cannels were measured (Figure 5 b) as suggested by one of the reviewers.

  1. Lines 300-303 need revision for clearing the statement.

Cleared as

Such studies are also useful at an earlier stage of the selection of a candidate drug among peptides. In vitro cell experiments are a mandatory part of preclinical studies of any candidate drug. The initial selection and study of peptide blockers is often performed in buffer solutions (for example, using electrophysiological and radioligand analysis), and the results need to be verified in a more complex and natural environment, such as a cellular one, where various nonspecific interactions and background proteolytic activity can affect the blocker-channel interactions.

  1. In lines 309-311, write a bit more detail regarding the role of Kv1.3 channel expression in some other ailments too.

A sentence describing a role of Kv1.3 channel in autoimmune diseases is added. 

  1. Line 333, the conclusion section should be explained in a good manner.

Done.

  1. In line 353, the word “elsewhere” is useless. Also replace it in other places where it is used incorrectly for citation.

Changed to “earlier” everywhere.

  1. In lines 364-365, for “different concentrations of peptide AgTx2 or KTx1 (0.04-100 nM) or crude venoms (40 mg/L) of scorpions M. eupeus” clarify the statement in a more precise way as it seems ambiguous either it is treatment with peptide or venom.

Corrected as

Displacement of AgTx2-GFP from the complexes with KcsA-Kv1.x (x=1,3,6) channels was analyzed by incubating spheroplasts with the constant concentration of AgTx2-GFP and increasing concentrations of peptide AgTx2 or KTx1 (0.04-100 nM) at 37°C for 2 h.  In similar experiments with scorpion venoms, AgTx2-GFP was added to spheroplasts together with a crude venom (40 mg/L) of scorpion M. eupeus or scorpion O. scrobiculosus.

  1. In lines 402-408 (Two-electrode….. Bessel filter.), please rephrase the statements in a concise manner, because they are not suitable for reader to apprehend

The statements are written in a similar way as we have done for more than 300 published papers in peer reviewed journals. Moreover, it is written comparable to ample research papers dealing with the same technique. The purpose of this section is to provide all the necessary information of our experimental setup in order for other researchers to be able to perform experiments in the same conditions. Therefore, the technical information is provided in this Material & Methods section. We sincerely do not know how to write this in a more concise manner.

Round 2

Reviewer 3 Report

Good for publication